# Geometry Analysis and Microhardness Prediction of Nickel-Based Laser Cladding Layer on the Surface of H13 Steel

**Fangping Yao [1,2,\*], Lijin Fang [3] and Xiang Chen [4]**

1   School of Mechanical Engineering and Automation, Northeastern University, Shenyang 110819, China
2   School of Mechanical Engineering and Automation, Liaoning University of Technology, Jinzhou 121001, China
3   Faculty of Robot Science and Engineering, Northeastern University, Shenyang 110819, China; jixie2009@foxmail.com
4   Engineering Training Center, Liaoning University of Technology, Jinzhou 121001, China; lnut21@163.com
*   Correspondence: sk4199642@163.com; Tel.: +86-134-6462-6919

**Abstract:** In order to improve the resistance to thermal fatigue and wear resistance of H13 hot-working tool steel, a nickel-based composite coating was prepared on its surface by laser cladding technology. The relationship was studied between the main processing parameters and the size of the cladding layer such as height and width. Based on the orthogonal polynomial regression method, the relationships were modeled mathematically between laser power, scanning speed, powder feeding voltage and microhardness. This model was proved to be able to predict the laser power and powder feeding voltage under 1100 Hv microhardness.

**Keywords:** laser cladding; nickel-based power; H13; microhardness; orthogonal polynomial regression method

## 1. Introduction

H13 steel is widely used for hot forging, hot-extrusion and die-casting [1–3]. Its surface is subjected to very severe conditions of cyclical thermal and mechanical load, and chemical and mechanical wear [4]. They cause thermal fatigue, corrosion, stress corrosion and welding on the surface of H13 steel, and service life is reduced [5]. Various surface treatment techniques are used to create a continuous compound layer and decrease the damage of crack, corrosion and erosion, such as quenching, physical vapour deposition, chemical vapor deposition process, laser cladding, etc. [4,6,7].

As one of the surface modification techniques, laser cladding provides thick protective coatings with a high quality on substrates. It is clean, efficient and has high bonding strength compared with others of thermal expansion and the melting point of the cladding materials [8,9]. According to processing requirements, laser cladding can clad a variety of metal powders selectively cladding small areas [10]. Self-fluxing powders are used commonly in laser cladding such as Ni-based, Co-based and Fe-based mixing powder [11]. Ni-based alloy is widely used to improve the hardness, wear resistance and corrosion resistance of parts at room temperature and high temperature [12–14].

Many researchers have carried out research studies on the geometric morphology and structure properties of laser cladding for different materials, and analyzed their relationship with laser cladding process parameters, in order to obtain accurate mathematical models for better application in production and processing.

Fallah, V. [15] developed a transient finite element model to simulate the temporal evolution of the melt-pool morphology and dimensions, and the geometry of the deposited material is finely predicted without assuming any of the geometrical characteristics, such as height, width, etc. The model presents a high accuracy in simulation of 3D melt-pool morphology and geometry in deposition of Ti45Nb on Ti–6Al–4V. Kai-yun Lei [16] built

a machine vision system to directly detect and measure the melt-pool morphology in wide-track laser cladding by high-power diode laser with a rectangle beam spot and it indicated that the width, length and area increased with increasing laser power, while increasing scanning speed caused the decreasing of the melt pool size.

Bourahima [17] investigated the impact of the process parameters on the coating geometry by using the method on analysis of variance, and it indicated the influence of each process parameter on the coating geometry (width, height) and the bonding quality. Chen [18] designed multi-track cladding experiments with L 27(313) orthogonal array by Taguchi method, and a prediction model based on support vector machine (SVM) was developed for the quality characteristics of the cladding coatings. GOODARZI [19] used statistical analysis method to model the relationship between the laser cladding process parameters and the clad layer geometry. CAIAZZO [20] used the artificial neural networks to find the correlation between the laser metal deposition process parameters and the output geometrical parameters of the deposited metal trace produced by laser direct metal deposition on 5 mm thick 2024 aluminum alloy plates.

Zhu [21] analyzed the effect of process parameters on surface smoothness in laser cladding, and the results indicated that the surface smoothness decreased with the increase of laser power and powder mass flow rate and the reducing of scanning speed, the degree of surface smoothness firstly increased then decreased with the growth of carrier gas flow rates. Yu [22] used the gray correlation degree to analyze the correlation between the quality of the cladding layer and the ideal cladding layer quality under different parameter combinations, and the results show that the laser power and scanning speed are the main factors affecting the morphology of the cladding layer. FAN [23] studied the optimal parameters for the laser cladding process by taking the hardness and dilution rate of the coatings as comprehensive indexes, and the mathematical model of the relationship was established by regression analysis between the geometrical size of the cladding layers with the process parameters.

Yadroitsev [24] used Greco-Latin square design to control geometrical characteristics of the tracks, and analysis of variance (ANOVA) was used to establish a statistically significant influence of the selective laser melting SLM process parameters on geometry of the single laser-melted track. The behavior of individual tracks and their geometric characteristics depend on the process parameters, and physical–chemical and granulo–morpho–metrical properties of the powder. Li [25] studied micro-segregation of laser clad IN718 super-alloy coatings fabricated under different atmosphere conditions, and the results indicated that the type of carrier gas had certain effects on the morphology and microstructure of the cladding layer, while type of shielding gas had no obvious effect. Huang [26] used artificial neural network (ANN) to predict the laser cladding parameters and the characteristic and performance of laser cladding layers, and the input parameters consisted of laser power, scanning velocity, laser spot diameter, and coating proportion, and the output parameters included the clad hardness, the clad width, and the clad height.

In the above studies, researchers used various algorithms to model the relationship between laser processing parameters and morphology, but they seldom modeled the microstructure and microhardness of the cladding layer. This paper analyzed the relationship between the laser processing parameters and the morphology of the nickel-based cladding layer on the surface of H13 steel, and it mathematically modeled the relationship between the laser processing parameters and the microhardness. It used the microhardness as the input parameter to predict the reasonable range of processing parameters.

## 2. Materials and Methods

In this research, the main components of nickel-based powder are shown in Table 1. Before doing the experiment, the powder was dried for at least 24 h, in order to reduce the composition changes caused by powder oxidation and ensure that the powder cannot stick to the inner wall of the powder feeding tube during the powder feeding process. The adhesion of the tube wall will affect the powder rate. The substrate is H13 steel in

the experiment, and its main components are shown in Table 2. In the experiment, the substrate is 100 mm × 50 mm × 10 mm, and the surface should be rubbed smooth with sandpaper to reduce the rust and impurities on the surface. In Figure 1, YLS-3000 fiber laser and KUKA robot are used in the experiment. Ar is used as the powder feeding gas and the protective gas in the laser cladding process to prevent the oxidation of molten pool elements due to high temperature. The laser spot diameter was 2 mm, and the defocus was 15 mm. Coaxial powder feeding method is adopted. The particle size of nickel-based alloy powder is 200 mesh.

**Table 1.** Main components of powder (%).

| Element | C | Cr | Si | B | Fe | WC | Ni |
|---|---|---|---|---|---|---|---|
| Value | 0.4–0.9 | 13–17 | 3.2–4.8 | 2.5–4.0 | ≤8.0 | 8.0 | Bal. |

**Table 2.** Main components of substrate (%).

| Element | C | Si | Mn | Cr | Mo | V | P | S | Fe |
|---|---|---|---|---|---|---|---|---|---|
| Value | 0.32–0.45 | 0.8–1.2 | 0.2–0.5 | 4.75–5.5 | 1.1–1.75 | 0.8–1.2 | ≤0.03 | ≤0.03 | Bal. |

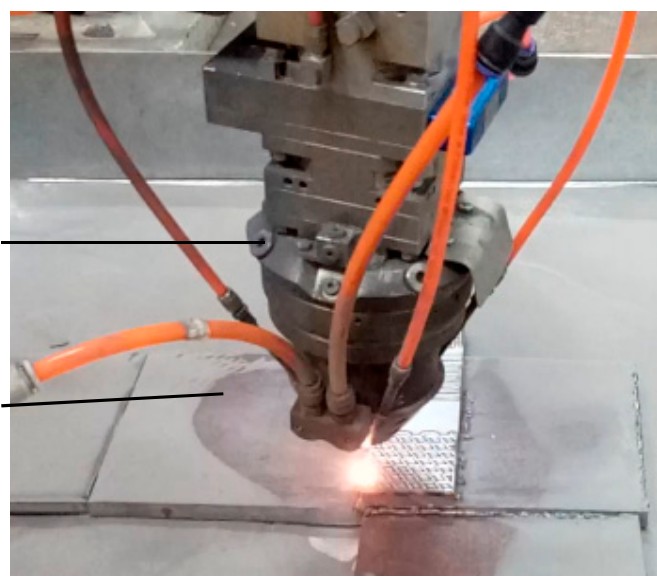

**Figure 1.** Single pass laser cladding experiment.

Multi-pass cladding should not be used to clad the turning tool edge if single-pass cladding can meet the processing. Multi-pass cladding not only consumes more material, but also damages tool substrate and cladding layer easily due to long-time laser radiation. This experiment designed an orthogonal test with 3 factors and 3 levels, and the levels of each factor are shown in Table 3.

**Table 3.** Orthogonal experimental design of process parameters.

| No. | Feeding Voltage F/V | Laser Power P/kW | Scanning Speed Vs/mm·s$^{-1}$ |
|---|---|---|---|
| 1 | $F_1 = 12$ | $P_1 = 0.9$ | $V_1 = 1$ |
| 2 | $F_2 = 14$ | $P_2 = 1.1$ | $V_2 = 2$ |
| 3 | $F_3 = 16$ | $P_3 = 1.3$ | $V_3 = 3$ |

After cladding, the cladding specimen was cut by wire cutting along the direction perpendicular to scanning, and the surface oxide scale was removed by sandpaper. The section was polished by MP-2B polishing machine, and it was etched by a mixture of nitric acid and hydrofluoric acid. The microhardness of the section of the laser cladding layer was tested by HV-1000 Micro Hardness Tester.

## 3. Results

### 3.1. Effect of Process Parameters on the Morphology of Cladding Layer

After the cladding experiment, the cladding width W and the cladding height H were measured. As shown in Figure 2, the microhardness was tested every 0.2 mm from the top of the cladding layer to the H13 steel matrix along the direction perpendicular to the cladding section. In practical application, the part lower than the substrate plane is not processed, so the microhardness is calculated in the cladding layer higher than the substrate surface, and the average value is taken as the microhardness value under each cladding parameter according to the measured value of each point and the number of points taken.

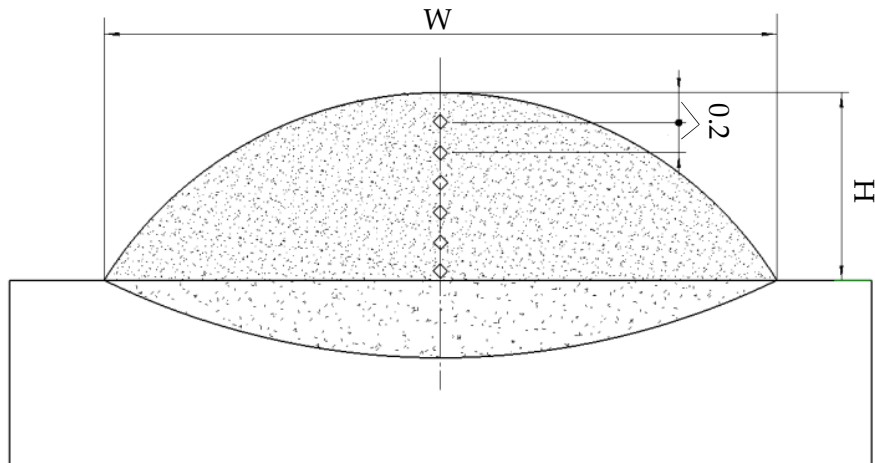

**Figure 2.** Microhardness measurement points.

As shown in Table 4, scanning speed is the most important factor for both the width and height of cladding. When the scanning speed increases, the laser energy irradiated on the cladding layer decreases relatively. As a result, less powder is melted, so the width and height of cladding are relatively reduced.

In the three parameters, the scanning speed has the greatest influence on the cladding width, followed by the laser power and the powder feeding voltage. For cladding height, scanning speed has the greatest influence, powder feeding voltage is the second, and laser power has the least influence. The optimal process combination is $C_1B_3A_2$, the powder feeding voltage is 14 V, the laser power is 1.3 kW, and the scanning speed is 1 mm/s. It is 6#, as shown in Figure 3b.

### 3.2. Modeling and Prediction of Microhardness of Cladding Layer

3.2.1. Effect of Process Parameters on the Microhardness of Cladding Layer

According to the microhardness analysis, the influence of the powder feeding voltage and laser power on the hardness is greater than the scanning speed, and the powder feeding voltage range is slightly greater than the laser power. The difference between scanning speed R value and other parameters is too large, and it indicates that this factor has little effect on microhardness.

**Table 4.** Orthogonal experimental table and results.

| No. | | Feeding Voltage (V) A | Laser Power (kW) B | Scanning Speed (mm/s) C | Cladding Width W (mm) | Test Parameters Cladding Height H (mm) | Microhardness D (Hv) |
|---|---|---|---|---|---|---|---|
| 1 | | 1 (12) | 1 (0.9) | 1 (1) | 3.23 | 1.03 | 863 |
| 2 | | 1 | 2 (1.1) | 2 (2) | 3.27 | 0.91 | 1080 |
| 3 | | 1 | 3 (1.3) | 3 (3) | 2.91 | 0.71 | 1017 |
| 4 | | 2 (14) | 1 | 2 | 2.99 | 0.91 | 1153 |
| 5 | | 2 | 2 | 3 | 2.83 | 0.67 | 1310 |
| 6 | | 2 | 3 | 1 | 4.89 | 1.39 | 1160 |
| 7 | | 3 (16) | 1 | 3 | 2.46 | 0.65 | 832 |
| 8 | | 3 | 2 | 1 | 3.73 | 0.97 | 1137 |
| 9 | | 3 | 3 | 2 | 3.67 | 0.85 | 904 |
| Cladding width W | I | 9.45 | 8.72 | 11.83 | Relationship of factors: C B A | | |
| | II | 10.69 | 9.81 | 9.93 | | | |
| | III | 9.86 | 11.47 | 8.21 | Best process parameter combination: | | |
| | R | 1.22 | 2.79 | 3.62 | $C_1 B_3 A_2$ | | |
| Cladding height H | I | 2.67 | 2.61 | 3.42 | Relationship of factors: C A B | | |
| | II | 2.98 | 2.59 | 2.64 | | | |
| | III | 2.51 | 2.97 | 2.03 | Best process parameter combination: | | |
| | R | 0.44 | 0.37 | 1.32 | $C_1 A_2 B_3$ | | |
| Microhardness D | I | −35 | −150 | 164 | Relationship of factors: A B C | | |
| | II | 626 | 531 | 141 | | | |
| | III | −125 | 86 | 160 | Best process parameter combination: | | |
| | R | 751 | 680 | 22 | $A_2 B_2 C_0$ | | |

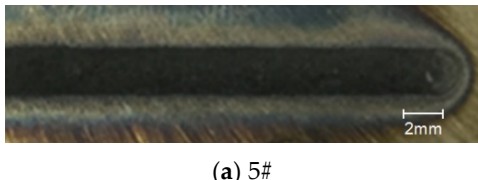

(**a**) 5#

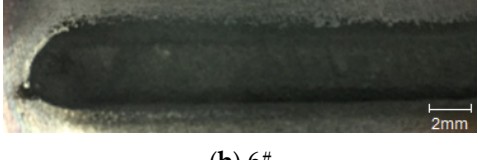

(**b**) 6#

**Figure 3.** Geometrical morphology of cladding layer.

The results show that the optimal process combination is $A_2B_2C_0$, where $C_0$ is within the value range. When the scanning speed was 3 mm/s, its microhardness is the maximum value. 5# is as shown in Figure 3a. Its average microhardness was 1311 Hv, and it is 1.5 times more than the substrate.

In the regression fitting of the mean microhardness, the microhardness in Table 3 is given in the form of ($y_i$-1000), as shown in Table 5. The orthogonal polynomial regression method is calculated as follows.

### 3.2.2. Orthogonal Polynomial Regression Modeling

Polynomial regression model [27,28]

$$y_i = \beta_0 + \beta_1 x_i + \beta_2 x_i^2 + \ldots + \beta_m x_i^m + \varepsilon_i, \ i = 1, \ 2, \ \ldots, n \tag{1}$$

Regression of y to x

$$y = \beta_0 P_0(x) + \beta_1 P_1(x) + \ldots + \beta_m P_m(x) + \varepsilon \tag{2}$$

**Table 5.** The orthogonal test table and experimental result.

| No. | | Feeding Voltage (V) | Laser Power (kW) | Scanning Speed (mm/s) | | Microhardness $y_i$-1000 (Hv) |
| :---: | :---: | :---: | :---: | :---: | :---: | :---: |
| | | A | B | C | Deviation | |
| 1 | | 1 (12) | 1 (0.9) | 1 (1) | 1 | −137 |
| 2 | | 1 | 2 (1.1) | 2 (2) | 2 | 80 |
| 3 | | 1 | 3 (1.3) | 3 (3) | 3 | 17 |
| 4 | | 2 (14) | 1 | 2 | 3 | 153 |
| 5 | | 2 | 2 | 3 | 1 | 310 |
| 6 | | 2 | 3 | 1 | 2 | 160 |
| 7 | | 3 (16) | 1 | 3 | 2 | −168 |
| 8 | | 3 | 2 | 1 | 3 | 137 |
| 9 | | 3 | 3 | 2 | 1 | −96 |
| Microhardness | I | −36 | −151 | 163 | 79 | |
| | II | 626 | 530 | 140 | 74 | T = 465 |
| | III | −126 | 85 | 161 | 312 | |
| | R | 752 | 681 | 23 | | |

When designing the orthogonal experiment, for the quantitative factor A, if its value is equally spaced, the main effect can be decomposed into the sum of the first-order term and the second-order term of the orthogonal polynomial, i.e.,

$$\mu(A) = a_0 + a_1 P_1(A) + \ldots + a_{t_A-1} P_{t_A-1}(A) \tag{3}$$

It submits the i-th level AI of factor A into Equation (3), and the following formula can be obtained.

$$\hat{a_0} = \sum_{i=1}^{t_A} \mu(A_i) = 0, \tag{4}$$

$$\hat{a_j} = \frac{1}{l_A \sum\limits_{i=1}^{t_A} P_j^2(A_i)} \left[ \sum_{i=1}^{t_A} P_j(A_i) y_i \right], \; j = 1, 2, \ldots, t_A - 1, \tag{5}$$

$$S_{\hat{a_j}}^2 = \frac{1}{l_A \sum\limits_{i=1}^{t_A} P_j^2(A_i)} \left[ \sum_{i=1}^{t_A} P_j(A_i) y_i \right]^2, \; f_{\hat{a_j}} = 1 \tag{6}$$

where yi is the sum of the i-th level data corresponding to A, $l_A$ is the repetition times of each level, $\mu(A_i)$ is the i-th level effect of *A*.

(1) Effect function of powder feeding voltage A

Transform the three equidistant levels of factor A into standard equidistant points

$$A_i' = \frac{A_i + h_A - A_1}{h_A} = \frac{A_i - 10}{2} \; i = 1, 2, 3 \tag{7}$$

where is the horizontal spacing of factor A.

The effect function is assumed to be quadratic orthogonal polynomials

$$\mu(A') = a_0 + a_1 P_1(A') + a_2 P_2(A') \tag{8}$$

According to Equation (4), $\hat{a_0} = 0$

According to Equation (5), $\hat{a_1} = -15$, $\hat{a_2} = -236$

According to Equation (6), $S_{\hat{a_1}}^2 = 1349$, $S_{\hat{a_2}}^2 = 111078$

(2)   Effect function of laser power B

Transform the three equidistant levels of factor B into standard equidistant points

$$B_i' = \frac{B_i + h_B - B_1}{h_B} = \frac{B_i - 0.7}{0.2} \ i = \ 1, \ 2, \ 3 \tag{9}$$

The effect function is assumed to be quadratic orthogonal polynomials

$$\mu\left(B'\right) = b_0 + b_1 P_1\left(B'\right) + b_2 P_2\left(B'\right) \tag{10}$$

According to Equation (4), $\overset{\wedge}{b_0} = 0$

According to Equation (5), $\overset{\wedge}{b_1} = 39$, $\overset{\wedge}{b_2} = -188$

According to Equation (6), $S_{\overset{\wedge}{b_1}}^2 = 9283$, $S_{\overset{\wedge}{b_2}}^2 = 70438$

(3)   Effect function of scanning speed C

$$C_i' = C_i \ i = \ 1, \ 2, \ 3 \tag{11}$$

The effect function is assumed to be quadratic orthogonal polynomials

$$\mu\left(C'\right) = c_0 + c_1 P_1\left(C'\right) + c_2 P_2\left(C'\right) \tag{12}$$

According to Equation (4), $\overset{\wedge}{c_0} = 0$

According to Equation (5), $\overset{\wedge}{c_1} = -0.4$, $\overset{\wedge}{c_2} = -7$

According to Equation (6), $S_{\overset{\wedge}{c_1}}^2 = 0.7$, $S_{\overset{\wedge}{c_2}}^2 = 108$

The coefficients of each effect function and their sum of squares of deviation are obtained, and the results are shown in Table 6.

Table 6. The coefficient of the effect function and the sum of its dispersion squared.

| Feeding Voltage F | | Laser Power P | | Scanning Speed Vs | |
|---|---|---|---|---|---|
| Effect function coefficient | Sum of squares of the deviation | Effect function coefficient | Sum of squares of the deviation | Effect function coefficient | Sum of squares of the deviation |
| $\overset{\wedge}{a_0} = 0$ | | $\overset{\wedge}{b_0} = 0$ | | $\overset{\wedge}{c_0} = 0$ | |
| $\overset{\wedge}{a_1} = -15$ | $S_{\overset{\wedge}{a_1}}^2 = 1349$ | $\overset{\wedge}{b_1} = 39$ | $S_{\overset{\wedge}{b_1}}^2 = 9283$ | $\overset{\wedge}{c_1} = -0.4$ | $S_{\overset{\wedge}{c_1}}^2 = 0.7$ |
| $\overset{\wedge}{a_2} = -236$ | $S_{\overset{\wedge}{a_2}}^2 = 111078$ | $\overset{\wedge}{b_2} = -188$ | $S_{\overset{\wedge}{b_2}}^2 = 70438$ | $\overset{\wedge}{c_2} = -7$ | $S_{\overset{\wedge}{c_2}}^2 = 108$ |

According to column D in Table 4, the sum of squares of error deviation $S_4^2$ and degrees of freedom $f_4$ are calculated:

$$S_4^2 = \frac{79^2 + 74^2 + 312^2}{3} - \frac{465^2}{9} \approx 12329 \tag{13}$$

$$f_4 = 3 - 1 = 2 \tag{14}$$

The sum of squares of the square deviation is as follows.

$$\frac{S_4^2}{f_4} = \frac{12329}{2} = 6164 \tag{15}$$

$S^2_{\hat{a}_1}$, $S^2_{\hat{c}_1}$ and $S^2_{\hat{c}_2}$ are all smaller than $S^2_4/f_4$. These terms are incorporated into the error term $S^2_e$. Finally, its value and degree of freedom can be respectively obtained:

$$S^2_e = S^2_4 + S^2_{\hat{a}_1} + S^2_{\hat{c}_1} + S^2_{\hat{c}_2} = 12,421.6 \tag{16}$$

$$f_e = 2 + 1 + 1 + 1 = 5 \tag{17}$$

The results of variance analysis are shown in Table 7.

$$F_{0.05}(1,5) = 6.61, \ F_{0.01}(1,5) = 16.3$$

**Table 7.** The analysis of variance.

| Sources of Variance | Variable Sum of Squares | Degrees of Freedom | Average Variable Sum of Squares | F | Significance |
|---|---|---|---|---|---|
| $\hat{a}_2$ | 111,078 | 1 | 111,078 | 41 | 0.5 |
| $\hat{b}_1$ | 9283 | 1 | 9283 | 3 | |
| $\hat{b}_2$ | 70,438 | 1 | 70,438 | 26 | 0.5 |
| error | 13,685 | 5 | 2737 | | |

Finally, the effect functions of powder feeding voltage (factor A) and laser power (factor B) are respectively obtained.

$$\mu(A) = -236 \times \left(\frac{A-14}{2}\right)^2 + 157 \tag{18}$$

$$\mu(B) = -188 \times \left(\frac{10B-11}{2}\right)^2 + 124 \tag{19}$$

The polynomial regression equation is as follows.

$$\hat{y} = \bar{y} + \mu(A) + \mu(B) = -236 \times \left(\frac{A-14}{2}\right)^2 - 188 \times \left(\frac{10B-11}{2}\right)^2 + 333 \tag{20}$$

where $\bar{y}$ is the total mean.

$$\bar{y} = T/n = 464/9 \approx 52 \tag{21}$$

where $T$ is the sum and $n$ is the number of data.

According to the significance analysis, the scanning speed (factor C) can be approximated as 0. In Table 6, the optimal condition is $A_2B_2C_0$, which is consistent with the results of range analysis. The predicted value of microhardness under the optimal production condition is as follows.

$$\hat{y_{opt}} = \bar{y} + \mu(A_2) + \mu(B_2) = 333 \tag{22}$$

The forecast error is as follows.

$$2S_y = 2 \times \sqrt{\frac{7573+9283}{5+1}} \approx 106 \tag{23}$$

The actual microhardness formula is as follows.

$$y_i = \hat{y} + 1000 \tag{24}$$

There is the comparison between calculated values and measured values from orthogonal multiple regression curves in Table 8.

**Table 8.** Comparison of the calculated value with the measured value.

| | $y_i$-1000 (Hv) | | | | | | | | |
|---|---|---|---|---|---|---|---|---|---|
| No. | 1 | 2 | 3 | 4 | 5 | 6 | 7 | 8 | 9 |
| Calculated value | −90.05 | 95.121 | −90.612 | 144.95 | 332.789 | 145.985 | −90.456 | 99.011 | −88.654 |
| measured value | −137 | 80 | 17 | 153 | 310 | 160 | −168 | 137 | −96 |

It can be seen from Table 7 that the third group of data has a big difference, but it is also within the range of error. It shows that the regression function can accurately reflect the relationship between the feeding voltage, laser power and the average microhardness.

### 3.2.3. Microhardness Prediction and Control

According to the requirements, the cladding layer should have a higher microhardness. Under the optimal hardness process parameters, the microhardness range can be predicted as follows.

When the powder feeding voltage is 14 V and the laser power is 1.1 kW, the calculated value of microhardness ($y_i$-1000) is estimated, $\hat{y_0} = 333$. When the significance level $\alpha$ is 0.05, the confidence interval for the microhardness $y_0$ is as follows.

$$\left[ \hat{y_0} - 2S_y < y_0 < \hat{y_0} + 2S_y \right] \tag{25}$$

It is [227, 439] The microhardness is basically at the maximum, and the microhardness is predicted to be between 1227 and 1439 Hv.

To obtain the microhardness above 1100 Hv, that is, the calculated value is above 100 Hv, and the powder feeding voltage (factor A) is selected between 12 and 16 V, and the laser power (factor B) is selected between 0.9 and 1.3 kW, and the powder feeding voltage is determined, the range of laser power control is

$$-236 \times \left(\frac{A-14}{2}\right)^2 - 188 \times \left(\frac{10B-11}{2}\right)^2 + 333 > 100 \tag{26}$$

Powder feeding voltage and laser power should be in the ellipse with boundary conditions. According to the above equation, the boundary conditions are as follows.

$$\begin{cases} B = 1.1 \quad A = 12.01 \text{ or } A = 15.99 \\ B \in \left[ \frac{11 - \sqrt{233 - 59 \times (A-14)^2}}{10}, \frac{11 + \sqrt{233 - 59 \times (A-14)^2}}{10} \right] \\ \quad 12.01 < A < 13.13 \text{ or } 14.87 < A < 15.99 \\ B \in [0.9, 1.3] \quad 13.13 \leq A \leq 14.87 \end{cases} \tag{27}$$

If the values of factor A and factor B are selected within the boundary condition area, there is generally more than 95% chance that the average microhardness is greater than 1100 Hv. The selected values of the powder feeding voltage (factor A) and laser power (factor B) are inside the curve, as shown in Figure 4.

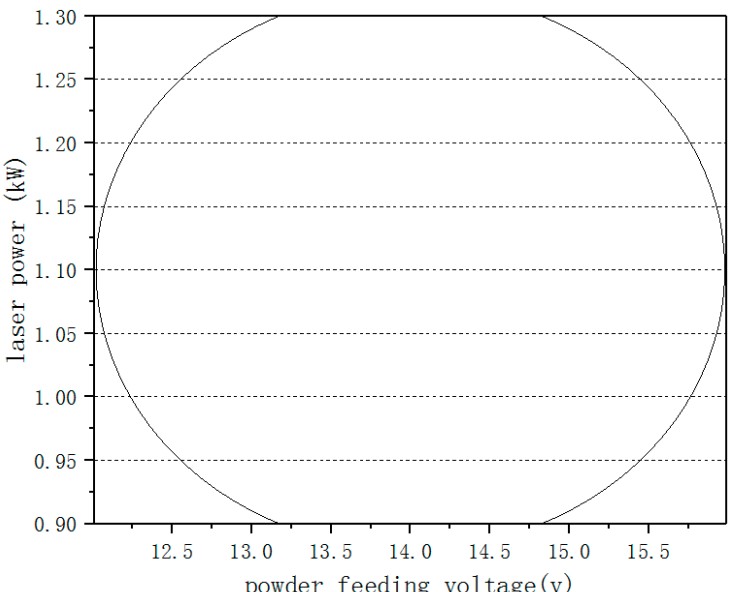

**Figure 4.** Range of laser power and powder feeding voltage.

## 4. Conclusions

In this paper, laser cladding technology was used to prepare a nickel-based coating on the surface of H13 steel. The surface morphology and microhardness of the cladding layer were tested and analyzed under different process parameters. Based on the results of this investigation, the following could be concluded from this work.

(1) The scanning speed is the most important factor for the width and height of the cladding layer, compared to the laser power and powder feeding voltage. The width and height of the cladding layer are relatively reduced when the scanning speed is gradually increased.

(2) A mathematical model can be obtained between the microhardness of the cladding layer and the laser power and powder feeding voltage through the nonlinear regression method. It can predict the numerical range of laser power and powder feeding voltage at a microhardness of 1100 Hv, with a confidence level of 0.95.

**Author Contributions:** F.Y. mainly designed, analyzed the data and wrote the manuscript; L.F. mainly directed this experiment; X.C. mainly performed the experiments. All authors have read and agreed to the published version of the manuscript.

**Funding:** This research was funded by Natural Science Foundation of China (No. 61803272) and Natural Science Foundation of Liaoning Province (No. 201602371).

**Institutional Review Board Statement:** Not applicable.

**Informed Consent Statement:** Not applicable.

**Data Availability Statement:** Data is contained within the article.

**Acknowledgments:** This research was funded by the Natural Science Foundation of China (No. 61803272).

**Conflicts of Interest:** The authors declare no conflict of interest.

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
