# Peer review of "Geometry Analysis and Microhardness Prediction of Nickel-Based Laser Cladding Layer on the Surface of H13 Steel"

_processes, doi:10.3390/pr9030408_

Round 1
Reviewer 1 Report
The article entitled "Geometry Analysis and Microhardness Prediction of Laser 2
Cladding Layer" sounds interesting. However some improvement needed before publishing
such as some related work cited in reference such as how laser cladding also affect the surface by annealing
Laser Annealing on the Surface Treatment of Thin Super Elastic NiTi Wire
S Samal, L Heller, J Brajer, O Tyc, L Kadrevek, P Sittner IOP Conference Series: Materials Science and Engineering 362 (1), 012007 how this process is better than thermal plasma technology spraying Thermal plasma technology: The prospective future in material processing Journal of cleaner production 142, 3131-3150 Finally author need to improve all the Figs 2-5, thats poorly readable. Its important to explain the highlight table , in bracket, its hard to understand. Why the figure is low quality and input marking is difficult to readable.Author Response
I revised the article, I submit the article to you as the attachment. Please see the attachment.

Reviewer 2 Report
Dear Authors,
Your article titled: “Geometry Analysis and Microhardness Prediction of Laser Cladding Layer” is aimed at assessing the influence of laser powder cladding process parameters on dimensions and hardness of beads. The problem was solved with the use of DOE. In my opinion, the work is valuable from a scientific and practical point of view. Below I present my remarks and comments in the order they appear in the text.
Title and Keywords: in my opinion, the sections of the article should contain information on materials: powder and substrate.
Abstract: correct the grammar. Abstract is too laconic. Please add a sentence of a general introduction, information about the material group to which H13 steel belongs (tool), and finally a sentence commenting the results. Change Hv on HV.
Introduction:
Throughout the work, please use the citation record in accordance with the journal guidelines: [1,2].
Line 39: what is "Wm, Lm and Am"?
Introduction has the wrong structure: only the first paragraph is written like review. The rest of the section is, in my opinion, simply listing works on a similar subject. Please rearrange the content of the lines: 32-84. In terms of laser cladding Introduction, it is well represented by references, however, there is no more general information about the use of a laser in the treatment of steels. In this regard, I can recommend the current article (not mine): https://doi.org/10.3390/ma13204540 and articles by Dr. Tomasz Kik and Prof. Aleksander Lisiecki from the Silesian University of Technology, Gliwice. This will also allow you to increase the number of cited articles and increase the recognition of the article.
Lines 85-87: target is not exactly described, material information missing: powder and substrate.
Lines 87-90: this section is premature: this is already a research description that should be in chapter 2.
M and M chapter:
This section starts… suddenly. The proper beginning of the description is missing.
Line 98: This sentence is not grammatically correct: "should be"?
Lines 119,120: change "corroded" to "etched". Did you use a mixture of these acids (etchant) or did you etching successively with both acids?
Table 4: What were the assessment criteria for determining: "better process parameter combination"? I guess it should be: "the best process ..."?
Line 145: correct "kW".
Chapter 3 uses a large number of equations. Please provide the source. What program was used for the calculations?
The tables are not properly numbered.
Table 6 caption: remove: "table".
Table 7 caption: replace: “Compare” with “Comparison”.
Conclusions: add an introductory sentence to conclusions. Line 289: "can be established" is too weak term in view of the results obtained. Correct V to HV in line 292.
Author Contributions: Format according to the journal's guidelines. There is no information on who wrote the manuscript.
Format the references according to the journal's guidelines. I believe that an article of this type should contain about 25 references. See also Processes, Materials, Metals journals, which publish papers on laser processes.
[2] the title of the journal is missing.
Please check the notation of commas and spaces, eg.: in lines 93, 138, 139, 155, 287, 295. Add spaces before the units. Correct the notation: "Table" (without a period). Use "Table" instead of "Tab.". Use "Figure" instead of "Fig.".
Reviewer 3 Report
In the current paper, authors have studied the geometry analysis and microhardness prediction of laser cladding layer. The paper lacks scientific analysis and discussion. It looks like reporting data without any comprehensive discussion. General comments that must be included to improve the value of paper:
Introduction section:
- General comment to introduction section: What is new in this work? Is this approach is new / novel? What makes authors work valuable in comparison to other work already published? There is no clear summary of introduction section. There is no information why nickel-based composite coating was used to improve the properties of H13 steel. No general justification of why such effects are studied etc.
Materials and methods section:
- Please clearly explain how the coating production conditions were determined.
- Initial particle size of powders should be emphasized.
- Line 120. How many hardness tests were performed? The load used in kgf was not mentioned as well.
Results and discussion section:
- Results obtained in this section and each subsection separately should be supported by sufficient references especially when ‘results’ and ‘discussion’ sections are combined into one. There are no references used in this section. The results obtained were not compared or supported by similar work. I feel that too much work was reported and not much discussed.
- Line 123. Effect of process parameters on the morphology of cladding layer were studied but was not clearly shown at all. The morphology of cladding layer was assessed based on Figure 3? I do not see clear justification.
- Line 133. Figure 2. Please improve the quality of the figure.
- Line 197. Figure 3. Please provide scale bar and enlarge the figures
- Line 133. Figure 4. Please improve the quality of the figure.
- Lines 155 – 157. “The powder feeding voltage and laser power together determine the effective utilization of laser and affect the grain refinement of cladding powder.” – any reference? Or SEM/OM pictures to support this statement?
- Table 4. What is the physical meaning of parameter D?
- Why such model was used to predict the microhardness distribution? I do not deny its correctness or effectiveness but it is hard to follow it.
- Line 280. Figure 7? Please doublecheck.
- Line 133. Figure 5. Please improve the quality of the figure.
- Maybe some comparison of hardness distribution along the cross section for both modelling and experimental studies should be included.
Conclusion section:
- Line 287-288. ‘However, the scanning speed is lower than the laser power and the powder feeding voltage for the influence of microhardness’ – cannot understand. Please rewrite.
The paper is interesting at some point but I strongly suggest to provide more scientific discussion.
Round 2
Reviewer 1 Report
The reviewer still maintain the previous comments.
Reviewer 2 Report
Dear Authors,
thank you for the answers. The article has been improved, but I believe it needs a few more improvements.
Change "Hv" on "HV"
Add spaces before the parentheses with references.
What computer program was used for the calculations?
Figure 4 is of poor quality.
Author contributions: The guidelines require just enter the authors' initials.
References were not properly formatted. Please check the journal's guidelines. Note the case of letters. As suggested in the first round of the review, I believe it would be beneficial for the visibility and future citation of the paper to cite the articles on laser welding and processing published in MDPI (for example: in Materials, Metals).
Reviewer 3 Report
No further comments. Minor suggestions:
Materials and methods section:
Point 2: Please clearly explain how the coating production conditions were determined. Response 1: This condition is already common in laser cladding, so no explanation is needed.
If such conditions are common then it should be mentioned in the text.
Point 3: Initial particle size of powders should be emphasized. Response 1: It has been added in the text:The particle size of nickel-based alloy powder is 200 mesh.
Initial particle size should be provided in µm.